METHODS AND RESOURCES

# Generation and validation of versatile inducible CRISPRi embryonic stem cell and mouse model

Rui Li[1], Xianyou Xia[1], Xing Wang[1], Xiaoyu Sun[1], Zhongye Dai[1], Dawei Huo[1,2], Huimin Zheng[3], Haiqing Xiong[4], Aibin He[4], Xudong Wu[1,2,5]*

1 State Key Laboratory of Experimental Hematology, The Province and Ministry Co-Sponsored Collaborative Innovation Center for Medical Epigenetics, Tianjin Key Laboratory of Cellular Homeostasis and Human Diseases, Department of Cell Biology, School of Basic Medical Sciences, Tianjin Medical University, Tianjin, China, 2 Department of Neurosurgery, Tianjin Medical University General Hospital, Tianjin, China, 3 Department of Prosthodontics, School and Hospital of Stomatology, Tianjin Medical University, Tianjin, China, 4 Beijing Key Laboratory of Cardiometabolic Molecular Medicine, Institute of Molecular Medicine, Peking-Tsinghua Center for Life Sciences, Peking University, Beijing, China, 5 Tianjin Key Laboratory of Epigenetics for Organ Development of Premature Infants, Tianjin, China

ʘ These authors contributed equally to this work.
* wuxudong@tmu.edu.cn

**Data Availability Statement:** All relevant data are within the paper and its Supporting Information files.

**Funding:** Funding: National key research and development program (grant number

## Abstract

Clustered regularly interspaced short palindromic repeat (CRISPR)-CRISPR-associated (Cas) 9 has been widely used far beyond genome editing. Fusions of deactivated Cas9 (dCas9) to transcription effectors enable interrogation of the epigenome and controlling of gene expression. However, the large transgene size of dCas9-fusion hinders its applications especially in somatic tissues. Here, we develop a robust CRISPR interference (CRISPRi) system by transgenic expression of doxycycline (Dox) inducible dCas9-KRAB in mouse embryonic stem cells (iKRAB ESC). After introduction of specific single-guide RNAs (sgRNAs), the induced dCas9-KRAB efficiently maintains gene inactivation, although it modestly down-regulates the expression of active genes. The proper timing of Dox addition during cell differentiation or reprogramming allows us to study or screen spatiotemporally activated promoters or enhancers and thereby the gene functions. Furthermore, taking the ESC for blastocyst injection, we generate an iKRAB knock-in (KI) mouse model that enables the shutdown of gene expression and loss-of-function (LOF) studies ex vivo and in vivo by a simple transduction of gRNAs. Thus, our inducible CRISPRi ESC line and KI mouse provide versatile and convenient platforms for functional interrogation and high-throughput screens of specific genes and potential regulatory elements in the setting of development or diseases.

## Introduction

Clustered regularly interspaced short palindromic repeat (CRISPR) and Cas (CRISPR-associated) proteins were originally found in bacteria and archaea to defend against viruses and plasmids by using CRISPR RNAs to guide the silencing of invading nucleic acids. Rapidly, this system has been simulated in other species by introducing the endonuclease Cas9 and single-

2017YFA0504102 to XWU). The funders had no role in study design, data collection and analysis, decision to publish, or preparation of the manuscript. National Natural Science Foundation of China (grant numer 81772676, 31970579 to XWU). The funders had no role in study design, data collection and analysis, decision to publish, or preparation of the manuscript. Natural Science Foundation of Tianjin Municipal Science and Technology Commission (grant number 18JCQNJC82300 to XWU). The funders had no role in study design, data collection and analysis, decision to publish, or preparation of the manuscript. Chinese Academy of Medical Sciences (grant number 157-Zk19-02 and Z20-04 to XWU). The funders had no role in study design, data collection and analysis, decision to publish, or preparation of the manuscript. Tianjin Medical University (Talent Excellence Program to XWU) The funders had no role in study design, data collection and analysis, decision to publish, or preparation of the manuscript.

**Competing interests:** The authors have declared that no competing interests exist.

**Abbreviations:** AAV, adeno-associated viral; ALP, alkaline phosphatase; BMT, bone marrow transplantation; Cas, CRISPR-associated; ChIP, chromatin immunoprecipitation; CRISPR, clustered regularly interspaced short palindromic repeat; CRISPRa, CRISPR activation; CRISPRi, CRISPR interference; dCas9, deactivated Cas9; DE, distal enhancer; Dox, doxycycline; EB, embryoid body; ESC, embryonic stem cell; FBS, fetal bovine serum; GFP, green fluorescent protein; GMEM, Glasgow's Minimum Essential Medium; GWAS, genome-wide association studies; ICE, inducible cassette exchange; ICM, inner cell mass, IF, immunofluorescence; KI, knock-in; KRAB, Krüppel-associated box; KRAB-ZFPs, KRAB containing zinc-finger proteins; LIF, leukemia inhibitory factor; LOF, loss-of-function; MOI, multiplicity of infection; MSC, mesenchymal stem cells; NGS, next generation sequencing; NPC, neural progenitor cell; OCT, optimum cutting temperature; PDLSC, periodontal ligament stem cell; PE, proximal enhancer; RT-qPCR, reverse transcription PCR; rtTA, reverse tetracycline transcriptional activator; sgRNA, single-guide RNA; SID, SIN3-interacting domain; SNP, single nucleotide polymorphism; TFAM, mitochondrial transcription factor A; TRE, tetracyclin response element; TSS, transcription start site; WT, wild type.

guide RNAs (sgRNAs) to cleave and edit specific DNA sequences. Ever since, the (CRISPR)/Cas9 system has been widely used as a powerful tool for genome editing [1–4]. Meanwhile, the RNA-guided epigenome editing technologies based on endonuclease deactivated Cas9 (dCas9) with 2-point mutations (D10A, H841A) have been developed. By fusion of dCas9 with transcription activator or repressor, the CRISPR activation (CRISPRa) or CRISPR interference (CRISPRi) allows researchers to control the level of endogenous gene expression [5–9].

Conventional genome editing by CRISPR/Cas9 technology may result in divergent indels and generate differential genotypes. In comparison, CRISPRi techniques block gene transcription by introducing transcription repressors at a defined genomic locus, leaving DNA sequence intact. Krüppel-associated box (KRAB) domains are the most commonly used repressors [6,7,10]. Epigenetic studies have demonstrated that KRAB containing zinc-finger proteins (KRAB-ZFPs) facilitate silencing by recruiting the KRAB-associated protein KAP1, and, in turn, other epigenetic repressors such as SETDB1, EHMT2/G9A, LSD1, and NURD complex. Thus, the dCas9-KRAB fusion protein creates an inactive chromatin environment by removing active chromatin mark-like histone H3-acetylation and establishing heterochromatin-like chromatin mark H3 lysine 9 trimethylation (H3K9me3) [11–14].

Current CRISPRi systems are usually generated by ectopic expression of dCas9-KRAB and sgRNAs via viral transduction. However, the expression of bacterial Cas9 could elicit host responses, aberrant cellular functions, or even toxicity in mammalian tissues [15,16]. Considering these potentially detrimental effects, the lasting expression of Cas9 or dCas9 proteins is not preferred. Besides, controllable genetic manipulation is crucial for most of the biological studies. Hence, the inducible expression of dCas9-KRAB serves as a better choice. Here, we generated a transgenic mouse embryonic stem cell (ESC) line with doxycycline (Dox) inducible and reversible expressions of dCas9-KRAB (iKRAB ESC). With this line, a simple transduction of sgRNAs enables us to do any locus-specific loss-of-function (LOF) studies in ESC or differentiated cells by proper timing of Dox addition or withdrawal. Having tested multiple gRNAs targeting promoters or enhancers, we found that our iKRAB system could efficiently maintain gene inactivation and control cell fate transition. And high-throughput screens could be performed in iKRAB ESC-derived cells. To facilitate broader applications, we took advantage of the ESC line and generated knock-in (KI) mice, which allowed inducible LOF studies ex vivo and in vivo. These systems especially the animal models empower us for functional studies and potentially high-throughput screens of specific genes or *cis*-regulatory elements.

## Results

### Generation and characterization of iKRAB ESC line

To generate a robust and inducible CRISPRi system, we took advantage of an engineered inducible cassette exchange (ICE) system of mouse ESCs (A2Loxcre) with reverse tetracycline transcriptional activator (rtTA) inserted into the *Rosa26* locus and tetracyclin response element (TRE)-LoxP-Cre-LoxP-Δneo integrated at the housekeeping gene *Hprt* [17]. Transgenes integrated at the *Hprt* locus remain transcriptionally active in differentiated cell types as well as in ESC. First, we constructed a dCas9-KRAB fragment onto the p2Lox-FLAG vector, which contains the LoxP sites [18]. Then, we pretreated A2Loxcre cells with Dox for 16 h so that *Cre* is expressed and that the cells are competent for recombination. Upon transfection with the p2Lox-FLAG-dCas9-KRAB construct, homologous recombination was initiated at the LoxP locus, and genomic fragments coming from plasmids were integrated into the downstream of the TRE promoter. At the same time, the Δneo gene acquired a *PGK* promoter and a start codon and enabled us to select for the precise integration with G418 (Fig 1A). After around 10 days of selection, the resistant clones were picked and characterized by genotyping PCR

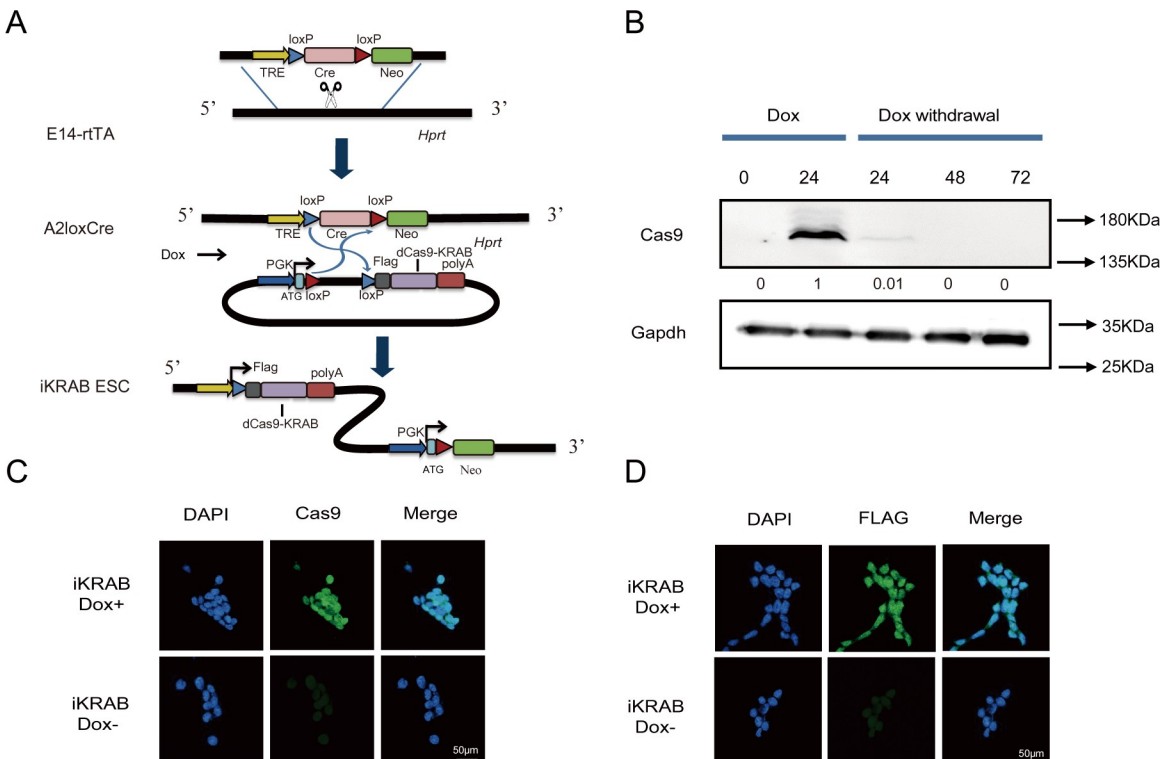

**Fig 1. Generation of the iKRAB ESC line.** (A) Schematic diagram shows the strategy of ICE to generate the iKRAB ESC line. FLAG-dCas9-KRAB was integrated into the downstream of the TRE element through homologous recombination. Dox-controlled rtTA drives the expression of fusion protein of FLAG-dCas9-KRAB. (B) Western blot analysis showing the inducible and reversible expression of FLAG-dCas9-KRAB protein at different time points after Dox addition or withdrawal. β-actin served as a loading control. A relative gray value quantification of dCas9-KRAB protein levels is below each lane of the band. (C, D) IF staining of Cas9 and FLAG in iKRAB ESC cultured with or without Dox. The scale bar represents 50 μm. Cas, CRISPR-associated; dCas9, deactivated Cas9; Dox, doxycycline; ESC, embryonic stem cell; ICE, inducible cassette exchange; IF, immunofluorescence; KRAB, Krüppel-associated box; rtTA, reverse tetracycline transcriptional activator; TRE, tetracyclin response element.

analysis. Two positive clones showed that the FLAG-dCas9-KRAB expressing sequence was precisely integrated downstream of TRE (S1A Fig). One of the clones was expanded for further analysis.

As examined by western blot assay with the Cas9 antibody, the clone did not express any detectable dCas9-KRAB protein when cultured without Dox, indicating no leaky expression. Upon addition of titrated concentration of Dox, dCas9-KRAB expression was robustly induced at 1 μg/ml after 24 h (S1B Fig and Fig 1B). Hereafter, we used Dox at 1 μg/ml for most of the experiments unless otherwise stated. Interestingly, the protein expression was gradually decreased to undetectable level 48 h after removing Dox (Fig 1B). Hence, we named this clone iKRAB ESC. Meanwhile, we did immunofluorescence (IF) analysis of the clone with Cas9 and FLAG antibodies. As shown in Fig 1C and 1D, the fusion protein was homogenously expressed after Dox treatment. These data showed that dCas9-KRAB expression could be precisely and reversibly controlled by the addition and withdrawal of Dox, which would allow controllable gene knockdown upon the introduction of gRNAs.

## CRISPRi efficiency at active genes

Many previous studies have demonstrated that dCas9-KRAB can achieve efficient knockdown of gene transcription, especially when tethered near the transcription start site (TSS) by

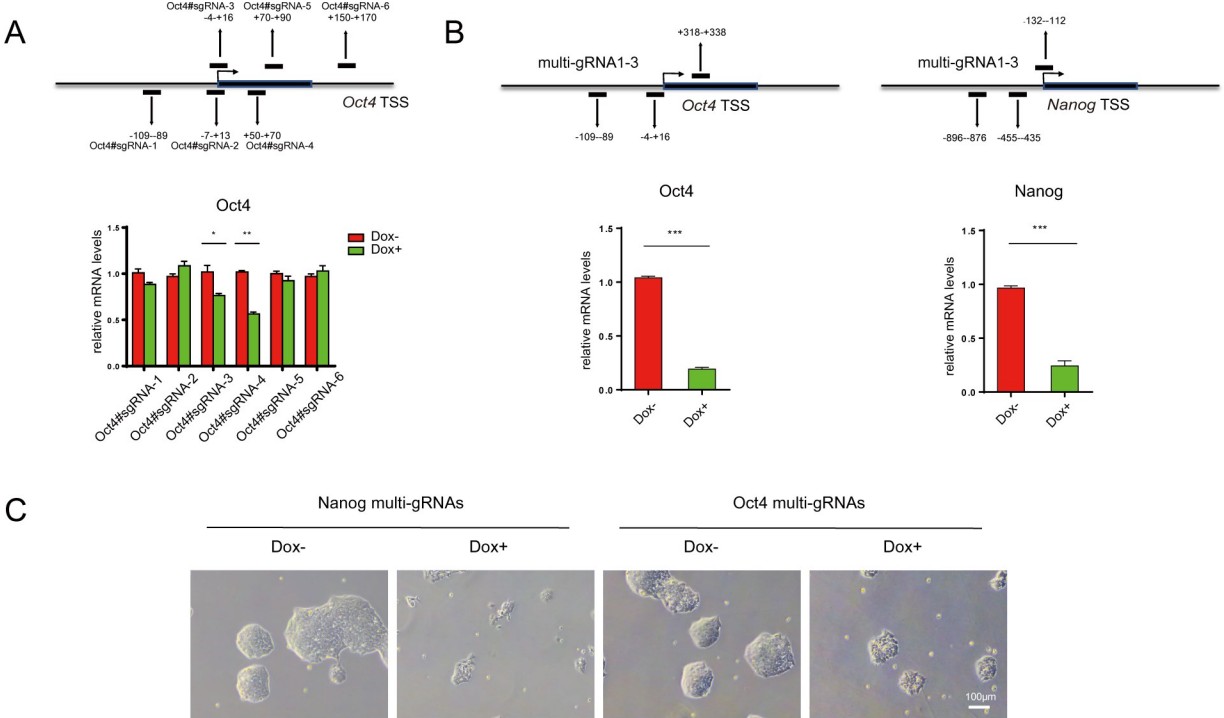

**Fig 2. Induced dCas9-KRAB at active gene promoters by sgRNAs or multi-gRNAs.** (A, B) RT-qPCR analysis of stable iKRAB ESCs containing sgRNA against *Oct4* or multi-gRNAs against Oct4 or Nanog after 2 days of Dox induction. The binding location of each gRNA is indicated relative to the TSS of *Oct4* or Nanog locus. (C) The cell morphology of Oct4 or Nanog multi-gRNAs-transduced iKRAB ESCs after 2 days of Dox induction. The scale bar represents 100 μm. The numerical values used to generate graphs in panels A and B are available in S1 Data. Cas, CRISPR-associated; dCas9, deactivated Cas9; Dox, doxycycline; ESC, embryonic stem cell; KRAB, Krüppel-associated box; RT-qPCR, reverse transcription PCR; sgRNA, single-guide RNA; TSS, transcription start site.

sgRNAs [6,8,9,19]. To test the efficiency of the iKRAB ESC line, we first designed 6 specific sgRNAs targeting near the TSS of *Oct4*. As Oct4 is 1 of the best known pluripotency factors and required for ESC self-renewal, its depletion is expected to result in a clear loss of pluripotent cell morphology. However, unexpectedly, quantitative reverse transcription PCR (RT-qPCR) analysis showed that none of the sgRNAs down-regulated the mRNA levels of Oct4 more than 50% after Dox induction, no matter targeting upstream or downstream of the TSS (Fig 2A). And IF analysis of cells transduced by Oct4#sgRNA4, the most effective one in our test, clearly showed that Oct4 expression was homogenously down-regulated after Dox treatment (S2A Fig). Thus, the inadequate down-regulation of Oct4 expression was not due to heterogeneous expression of sgRNAs or inadequate transduction rate. Therefore, our data argue against the high CRISPRi efficiency by sgRNA-guided dCas9-KRAB on actively transcribed genes.

To find out how the induced KRAB reconfigures chromatin at active versus inactive genes, we did chromatin immunoprecipitation (ChIP)-qPCR analysis in Oct4#sgRNA4 and Fgf5#sgRNA1-transduced iKRAB ESCs. FLAG or Cas9 ChIP-qPCR analysis demonstrated that the efficiency of tethering dCas9-KRAB fusion protein at the designed locus in either of the Dox-treated cells was comparable. It indicates that the genome binding of dCas9-KRAB protein was independent of gene transcription levels. Then, we did ChIP-qPCR analyses for the repressive histone mark H3K9me3. Interestingly, H3K9me3 levels are increased at the locus of *Oct4* as well as of *Fgf5*. However, H3K9me3 spreads more than 6 kb at *Fgf5* TSS, while it only spreads around 2 kb at *Oct4* TSS (S3A Fig). These data suggest that sgRNA-tethered

dCas9-KRAB at the active chromatin region is not sufficient to counteract active transcription and spread the heterochromatin-like state. Actually, it has been demonstrated that loss of activation signals usually precedes transcriptional repression and deposition of repressive chromatin marks [20–23]. Hence, it is not really surprising to observe inadequate CRISPRi at the presence of active transcription when using sgRNAs.

Based on the above regulatory mechanism, multi-gRNAs by linking multiple gRNAs linearly are supposed to induce more widespread heterochromatin-like state and thereby work more efficiently to suppress active gene transcription than sgRNAs. For this reason, we cloned multi-gRNAs targeting the TSS of *Oct4* and *Nanog* and transduced the iKRAB ESCs, respectively. Indeed, the multi-gRNAs targeting *Oct4* TSS led to more than 12 kb spreading of H3K9me3 mark (S3B Fig). And accordingly, these multi-gRNAs achieved much higher knockdown efficiency (above 75%) than sgRNAs (Fig 2B). The induced ESCs failed to self-renew and lost the normal morphology (Fig 2C). Therefore, multi-gRNAs are recommended to achieve high CRISPRi efficiency on actively transcribed genes.

Nevertheless, highly efficient gene inactivation is not always preferred. In some cases, different levels of gene down-regulation may help get insight into gene dosage effect. For instance, we observed that the ratio of Nanog-positive cells was even increased in the above SL-cultured Oct4-CRISPRi ESCs, suggesting a robust pluripotent state (S2A and S2B Fig). This is consistent with a previous unexpected finding that the self-renewal efficiency in Oct4 +/− ESCs is even enhanced compared with the wild type (WT) [24]. Thus, the iKRAB system may provide a valuable means to precisely control the dosage of target gene expression.

## iKRAB efficiently maintains gene inactivation

A main advantage of an inducible system is to precisely control gene expression via the timing of Dox addition. Since the iKRAB was not sufficient to induce gene inactivation, we followed to test whether it would maintain gene inactivation in dynamic settings. We took advantage of the switch between naïve and primed pluripotent states of iKRAB ESC. ESCs cultured in serum-free medium containing GSK3 inhibitor and MEK inhibitor (2i) and leukemia inhibitory factor (LIF) (hereafter 2i) supports naïve pluripotency that mimics the inner cell mass (ICM) of the blastocyst. Upon switch to serum-containing medium with LIF (hereafter SL) or Fgf2 and activin (hereafter FA), ESCs will exit from naïve pluripotency and switch to primed pluripotency. Some factors such as the epiblast maker Fgf5 and the histone methyltransferase Mll1 are switched on during the transition, accompanied with loss of Nanog expression [25–27] (Fig 3A). We transduced iKRAB ESCs with lentivirus expressing sgRNAs targeting the TSS of *Fgf5* or *Mll1*. In the absence of Dox, *Fgf5* or *Mll1* expression was strongly induced upon the switch of the culture condition from 2i to SL or FA. Although Dox addition in 2i medium exerted minor effects on *Fgf5* or *Mll1* expression, it strongly suppressed the induction by medium switch (>85%) (Fig 3B). Then, we further took the Fgf5#sgRNA2 transduced ESCs to observe how its CRISPRi affects pluripotent states. As shown by the cell morphology and immunostaining, the expression of Oct4 and Nanog are decreased to undetectable levels in the control (Dox-) cells upon switching from 2i to FA condition. In contrast, the Dox-treated cells maintained as round colonies, with majority of them Oct4 and Nanog-positive even in response to FA signals (Fig 3C and 3D). These data indicate that preset Fgf5 inactivation is maintained and thereby hinders the exit of naïve pluripotency.

In comparison with Dox addition in 2i medium, when we initiated Dox treatment at the primed state with *Mll1* highly expressed (FA culture), the same sgRNAs targeting the TSS of *Mll1* achieved moderate down-regulation efficiency (approximately 60% down-regulation) (Fig 3E). Nonetheless, upon being switched back to 2i medium, a subset of the Mll1

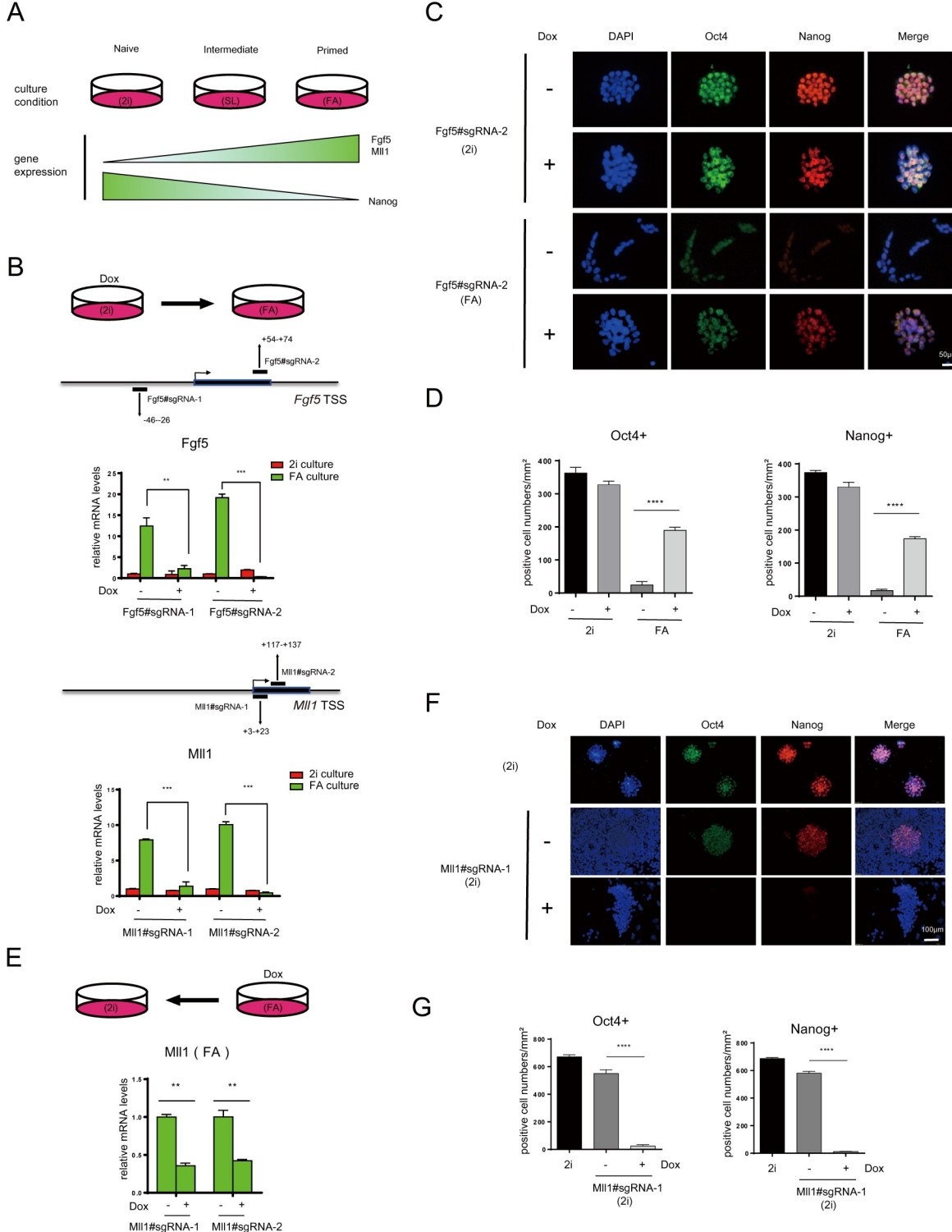

**Fig 3. Induced dCas9-KRAB at promoters is sufficient to maintain gene inactivation.** (A) Schematic diagram shows the dynamics of different states and marker gene expression of pluripotent stem cells cultured in different conditions. (B) RT-qPCR analysis of Fgf5 and Mll1 mRNA levels of iKRAB cells containing designated sgRNAs (with or without Dox) upon culture condition switch from 2i to FA. The binding location of each sgRNA is indicated relative to the TSS of the *Fgf5* or *Mll1* locus. (C) Representative IF staining of Oct4 and Nanog in iKRAB cells containing Klf5#sgRNA-2 treated with or without Dox in 2i condition or 2i switch to FA condition. The scale bar represents 100 μm. (D, G) The relative Oct4 and Nanog-positive cell numbers (normalized by DAPI+ cell numbers) are compared. (E) RT-qPCR analysis of Mll1 mRNA levels of iKRAB cells containing the same sgRNAs as (C) (with or without Dox) in FA condition. (F)

Representative IF staining of Oct4 and Nanog in iKRAB cells containing Mll1#sgRNA-1 treated with or without Dox for 4 days in FA condition, followed by switch back to 2i condition. ESC constantly cultured in 2i condition was used as a control. The scale bar represents 100 μm. Data in B, D, E, and G are represented as the mean ± SD of replicates (*n* = 3) (*$p < 0.05$, **$p < 0.01$, ***$p<0.001$, ****$p < 0.0001$; and 2-tailed unpaired *t* test). The numerical values used to generate graphs in panels B, D, E, and G are available in S1 Data. ESC, embryonic stem cell; Dox, doxycycline; dCas9, deactivated Cas9; IF, immunofluorescence; RT-qPCR reverse transcription PCR; SD, standard deviation; sgRNA, single-guide RNA.

knockdown cells regained Oct4 and Nanog expressions. In contrast, Oct4 and Nanog-positive cells could hardly be observed in the control group (Fig 3F and 3G). It indicates that Mll1 down-regulation facilitates the reprogramming from primed state to naïve state as previously reported [27]. This finding also prompts us for further CRISPRi screening to identify more chromatin regulators whose suppression may contribute to the reprogramming. Together, these data demonstrate that the iKRAB system is highly efficient to block gene activation, although inadequate to induce repression of active genes.

## iKRAB maintains inactive enhancers

Epigenome editing is supposed to modulate activities of any potential *cis*-acting regulatory elements. Enhancers are a vital regulatory element for tissue or development stage-specific gene expression through interaction with promoters. Thus, we tested how the iKRAB system works at enhancers in ESC and derived cells. An attractive model is the dynamic reorganization of enhancers between the 2 states of pluripotent stem cells [28,29]. For example, *Oct4* expression is controlled by different enhancers, distal enhancer (DE) in naïve state while proximal enhancer (PE) in primed state, although *Oct4* is expressed in both naïve and primed pluripotent cells [28,30]. When 2 sgRNAs targeting *Oct4* PE were respectively introduced into the 2i cultured iKRAB ESCs, Dox treatment induced no effects or even slight up-regulation of Oct4 expression, as shown by RT-qPCR analysis (Fig 4A and S4 Fig). Upon the culture condition switching from 2i to either SL or FA without Dox, Oct4 expression levels were increased in SL condition, an intermediate state with simultaneous activation of 2 enhancers. However, after Dox treatment, Oct4 activation was successfully suppressed in SL condition and was almost abrogated in FA condition (Fig 4B). IF analysis further confirmed that Oct4 expression was impeded in FA condition at the presence of Dox, although it was unaffected in 2i condition. Moreover, we found that Oct4 expression was restored 4 days after withdrawal of Dox, indicating CRISPRi effect as well as the expression of dCas9-KRAB fusion protein was reversible (Fig 4C and 4D). ChIP-qPCR analysis of 2i and FA-cultured cells showed that H3K27me3 and H3K9me3 levels at *Oct4* PE were decreased, together with increased H3K4me1 and H3K27ac levels upon switch from 2i to FA condition. However, when culture condition was switched at the presence of Dox, H3K27me3 and H3K9me3 levels at *Oct4* PE were maintained or even increased, and the increase of H3K4me1 and H3K27ac levels was hindered (S5 Fig). These data clearly indicated that *Oct4* PE was blocked at inactive state by induced dCas9-KRAB.

To further test the eligibility of iKRAB ESC for the dissection of specific enhancers during differentiation, we established a neural differentiation model to observe the dynamic control of *Sox2* enhancers and its downstream effects. Sox2 is highly expressed in neural progenitor cells (NPCs) as well as in ESCs; however, its expression is activated by DE in ESCs while likely by PE in NPCs [31,32]. We transduced the iKRAB ESCs with lentivirus expressing a specific sgRNA targeting *Sox2* PE before proceeding with embryoid body (EB) differentiation. When RA-induced NPC differentiation was initiated, cells were cultured with or without Dox (Fig 4E). RT-qPCR analysis in 8 days differentiated NPCs showed that Sox2 mRNA levels were significantly decreased in the Dox-treated group compared with the mock control, indicating that the switch-on of *Sox2* PE was blocked (Fig 4F). Consistent with the activation of *Sox2* PE

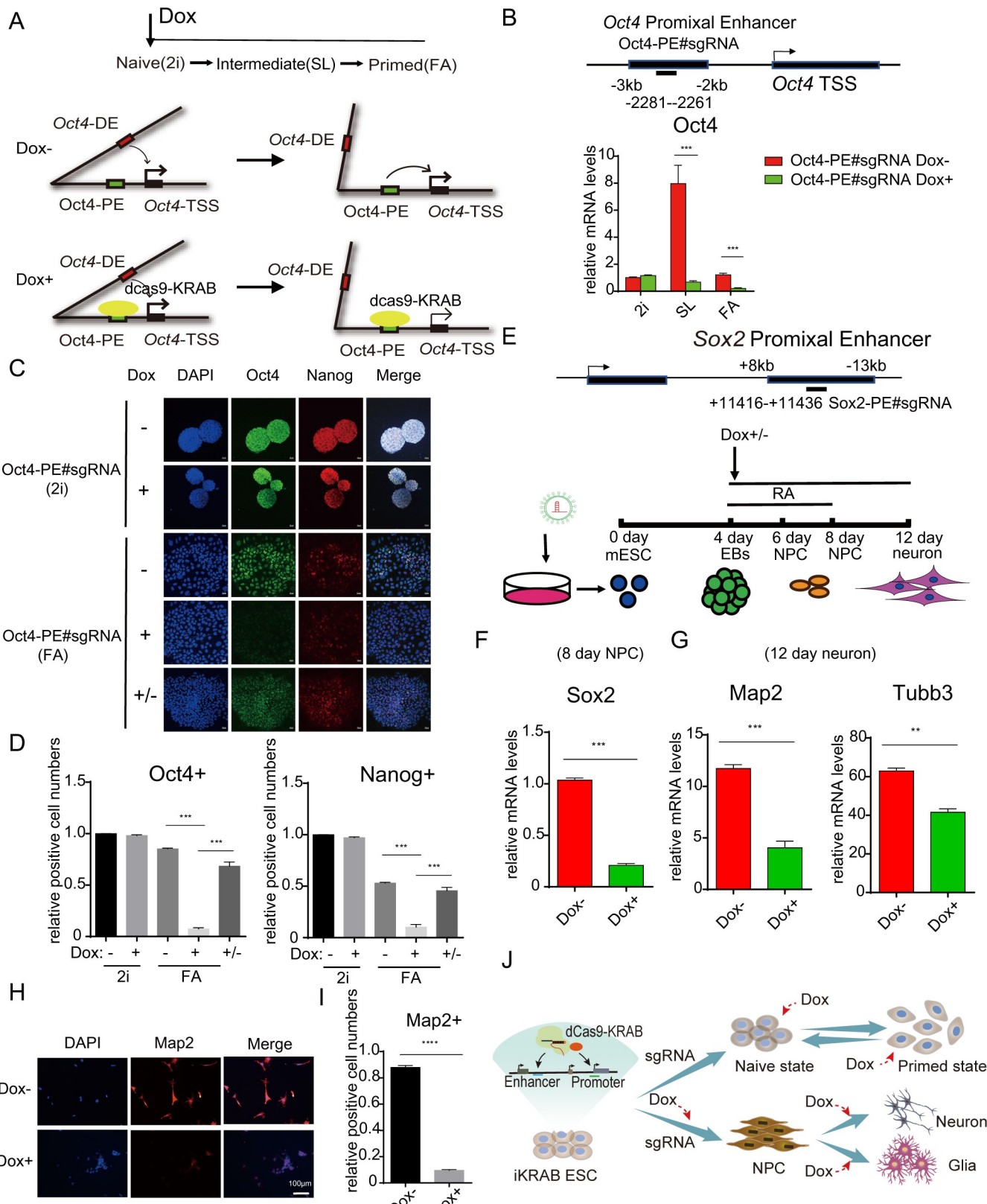

**Fig 4. Induced dCas9-KRAB at enhancers is sufficient to maintain gene inactivation in response to activation signals.** (A) Schematic diagram shows the PE of *Oct4* is activated upon culture condition switch from 2i to FA. Dox-induced dCas9-KRAB is tethered to the PE of *Oct4* in 2i condition, and Oct4

expression is to be tested in SL and FA conditions. (B) RT-qPCR analysis of Oct4 mRNA levels of iKRAB cells containing designated sgRNAs (with or without Dox) upon culture condition switch from 2i to SL or FA. The binding location of each sgRNA is indicated relative to the PE of *Oct4* locus. (C) Representative IF staining of Oct4 and Nanog in designated conditions. The scale bar represents 20 μm. (D, I) The relative Oct4, Nanog, or Map2-positive cell numbers are compared. (E) Schematic diagram shows iKRAB cells containing designated sgRNAs the PE of *Sox2* are differentiated to NPC and neuron. Dox was added before NPC stage together with RA. (F) RT-qPCR analysis of Sox2 mRNA levels in NPC (8 days) from the group with or without treatment. (G) RT-qPCR analysis of 2 neuron marker genes in neuron (12 days) from the group with or without treatment. (H) Representative IF staining of Map2 in neuron (12 days) from the group with or without treatment. The scale bar represents 50 μm. (J) A model for the temporal control of enhancer or promoter by the timing of Dox addition during ESC differentiation. Data in B, D, F, G, and I are represented as the mean ± SD of replicates ($n$ = 3 or 4) (**$p < 0.01$, ***$p < 0.001$, ****$p < 0.0001$; and 2-tailed unpaired $t$ test). The numerical values used to generate graphs in panel B, D, F, G, and I are available in S1 Data. Dox, doxycycline; dCas9, deactivated Cas9; IF, immunofluorescence; KRAB, Krüppel-associated box; NPC, neural progenitor cell; PE, proximal enhancer; RT-qPCR, reverse transcription PCR; sgRNA, single-guide RNA.

in NPCs, the chromatin interaction between *Sox2* PE and TSS was increased after RA treatment, as shown by the 3C-PCR analysis. And this interaction failed to be established in the Dox-treated cells (S6 Fig), suggesting that induced dCas9-KRAB suppressed *Sox2* PE activation and the associated chromatin looping. Then, we continued the differentiation of NPCs to neuron for each group and observed how Sox2 inactivation affected sequential neurogenesis. RT-PCR analysis showed that the expression levels of 2 neuron marker genes *Map2* and *Tubb3* were significantly lower in the Dox-treated group than the control group (Fig 4G). The inadequate Map2 expression in the Dox-treated group was further confirmed by IF analysis, indicating differentiation defects (Fig 4H and 4I). Collectively, a simple introduction of sgRNAs in iKRAB ESC can efficiently guide temporally induced dCas9-KRAB to block the activation of lineage-specific gene promoters or enhancers and thereby affect cell fate transitions (Fig 4J).

## iKRAB ESC for high-throughput screening

As ESCs have the potential to differentiate into all cell types of the organism, the iKRAB ESC is supposed to be optimal for the identification of specific promoters or enhancers and characterization of associated genes at defined contexts. And hence, we developed a CRISPRi screen to identify specific chromatin regulators whose inhibition would alleviate the toxicity of sodium channel blockers in the derived neural cells. First, we created a sgRNA library (containing 5,115 sgRNAs with 5,096 specific sgRNAs and 19 nontargeting negative control sgRNAs) that targeted the TSSs (including different transcripts) of known or potential epigenetic regulators and RNA binding proteins coding genes (total number = 857). Then, we transduced the sgRNA library into the iKRAB ESC, induced CRISPRi activity before differentiation to NPC stage while treated the cells with NaV1.8 channel blocker A803467. The Dox-free group of NPCs almost completely died after 4 days of A803467 treatment. The survived cells in Dox-treated group were harvested for deep sequencing (Fig 5A).

Compared with the input, 41 sgRNAs for 19 genes were significantly enriched in the survived cells (log10(Fold change Dox + /input) > 1, S1 Table). *Cirbp*, *Prmt2*, and *Dgkh* were among the top hits (Fig 5B). Then, we followed to validate certain hits by inducible shRNAs. We transduced ESCs with either of 2 inducible shRNA lentiviral constructs against Prmt2. After confirming the knockdowin efficiency by Dox induction (S7 Fig), we proceeded for NPC differentiation. As observed by Map2 IF analysis, induced Prmt2 depletion does not obviously affect the NPC differentiation in the control group (DMSO) (Fig 5C 2 upper panels and Fig 5D). Not surprisingly, A803467 negatively selected neural cells as rather few Map2-positive cells survived. However, the neural differentiation was rescued by induced Prmt2 depletion as majority of Map2-positive cells survived in the Dox-treated group (Fig 5C bottom panels and Fig 5D).

PRMT2 belongs to type I protein arginine methyltransferases, and there were inhibitors available against its activity, such as AMI-1. So we examined whether AMI-1 would alleviate

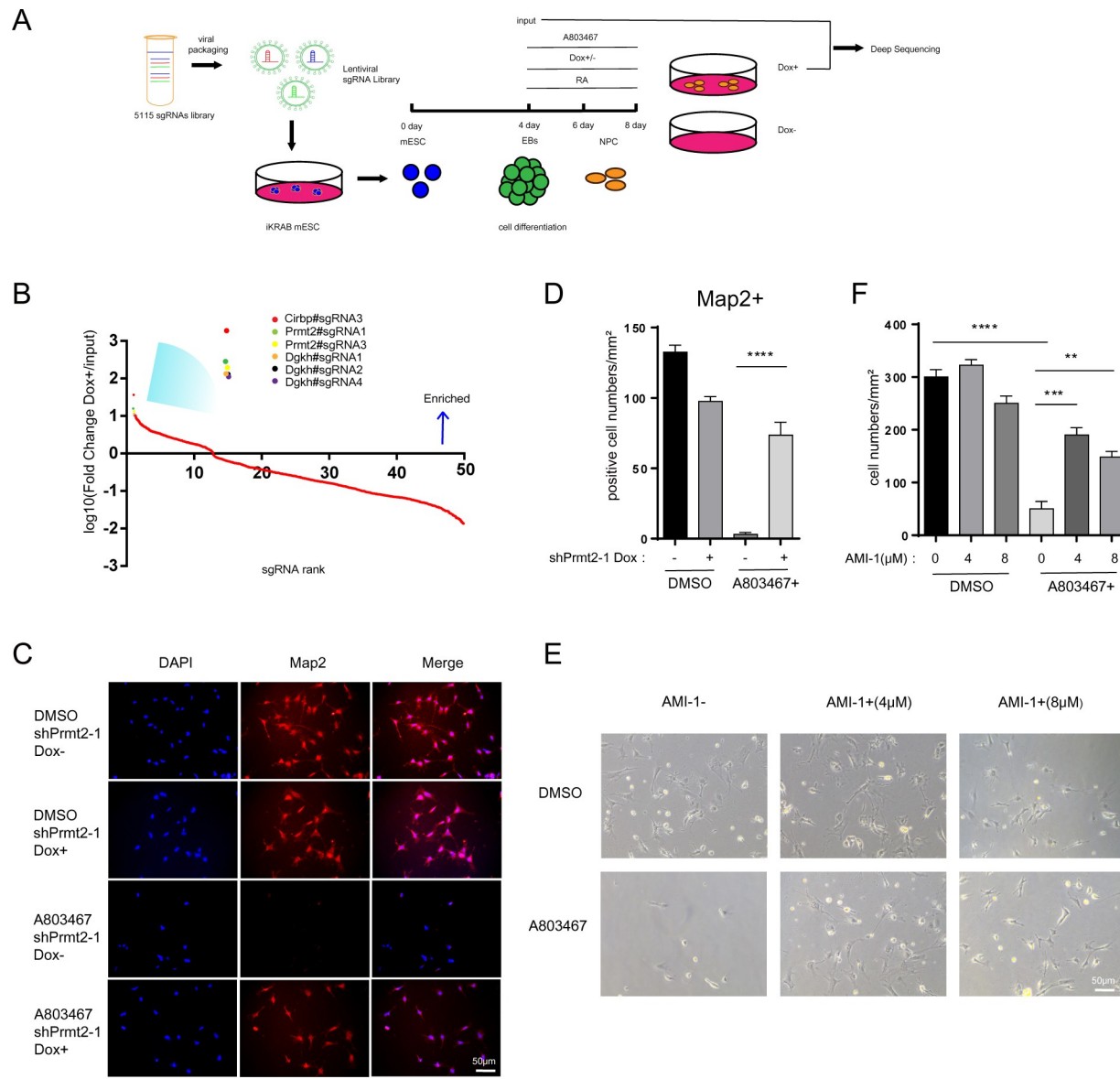

**Fig 5. iKRAB ESC for LOF screening.** (A) Schematic representation of CRISPRi screens in the iKRAB ESC-derived neural cells to evaluate chromatin regulators whose depletion would resist the toxicity of sodium channel blockers A803467. (B) A map of the contribution of the top 414 sgRNAs enriched in A803467-resistant cells. PRMT2 is among the top hits. (C) Representative IF staining of Map2 in designated groups. The scale bar represents 100 μm. (D) The relative Map2-positive cell numbers are compared. (E) Validation the neuroprotective effects of PRMT inhibitor AMI-1. The scale bars represent 50 μm. (F) The relative number of viable cells in each designated group (numbers per mm²) is compared. DMSO was used as a negative control in the assay. Data in D and F are represented as the mean ± SD of replicates (*n* = 3 or 4) (**p < 0.01, ***p < 0.001, ****p < 0.0001; and 2-tailed unpaired *t* test). The numerical values used to generate graphs in panel D and F are available in S1 Data. CRISPRi, CRISPR interference; ESC, embryonic stem cell; IF, immunofluorescence; LOF, loss-of-function; SD, standard deviation; sgRNA, single-guide RNA.

the toxicity of A803467 in NPCs. As shown in Fig 5E and 5F, few cells survived at the presence of A803467 alone. However, addition of AMI-1 significantly protected NPC from cell death and largely maintained the cell morphology at the presence of A803467. Accordingly, the CRISPRi screening identified toxicity resistance gene and provided possible solutions for neuroprotection. Thus, the iKRAB ESC may serve as a versatile model for LOF screens of functional genes or regulatory elements in a wide range of settings.

## Generation of an iKRAB KI mouse model for inducible gene silencing ex vivo and in vivo

Although dCas9-KRAB has been broadly applied in a few cell lines [8,9,33,34], a robust CRIS-PRi system for in vivo application is still urgently needed. Since we have confirmed the CRIS-PRi effects of the iKRAB ESC, we performed blastocyst injection. After successfully obtaining mouse chimeras and screening of germline transmitted offsprings, chimeric founder mice were crossed to generate iKRAB homozygous KI mice, verified by genotyping PCR (Fig 6A and 6B). The iKRAB KI mice were fertile, presented no morphological abnormalities, and were able to breed to homozygosity. Western blot assay of protein lysates from the mouse tails showed that no expression of dCas9-KRAB was observed in the KI mice until being fed with Dox-containing water (Fig 6C). Then, we tested inducible CRISPRi effect ex vivo and in vivo.

To test the CRISPRi effect ex vivo, we took advantage of a previously established osteoblast differentiation model of mesenchymal stem cells (MSC) [35]. Periodontal ligament stem cells (PDLSCs) were harvested from 8-week-old iKRAB KI mice. After a short expansion in MSC medium, the cells were switched to differentiation medium (Fig 6D). RT-qPCR analysis showed that Runx2 expression was activated after 5 days of differentiation (Fig 6E). Meanwhile, alkaline phosphatase (ALP) activity, an early marker for osteoblast differentiation, was strongly induced (Fig 6F). Then, we transduced PDLSCs with lentivirus expressing specific sgRNA targeting the TSS of *Runx2*, encoding a key transcription factor driving osteoblast differentiation. The transduced cells were then switched to differentiation medium with or without Dox. We found that Dox treatment modestly but significantly suppressed the differentiation medium-induced Runx2 activation (Fig 6E) and ALP activity (Fig 6F). These data illustrate that primary cells isolated from the iKRAB KI mice respond well to Dox induction.

To directly test the CRISPRi effect in vivo, we designed 3 sgRNAs, which respectively target the specific enhancer and the first exon of *TFAM* (*mitochondrial transcription factor A*), whose depletion results in muscle atrophy [36]. To achieve high knockdown efficiency in vivo, we cloned multi-gRNAs by linking the 3 sgRNAs linearly into an adeno-associated viral (AAV) vector simultaneously expressing green fluorescent protein (GFP). The construct was used for AAV packaging and purification. Then the high titer virus was injected into the *tibialis anterior* muscle of 6-week-old iKRAB KI mice. Dox-containing water was fed to the mice 2 weeks after injection. We chose to test in muscles mainly considering of technical feasibility and local virus concentration. Mice were humanely killed after 1 month of Dox induction, and tissues from the *tibialis anterior* muscle were isolated for analysis (Fig 7A). Successful transduction was confirmed by the observation of GFP. And Tfam expression levels were significantly lower in GFP+ cells than in the GFP− cells as shown by RT-qPCR analysis of the sorted cells (Fig 7B). IF analysis of GFP and laminin expression at the myofiber membrane demonstrated that the diameter of muscle fibers was significantly smaller in the AAV-infected muscle fibers (GFP+) compared with the uninfected ones (GFP−) (Fig 7C). The cross-sectional areas of the GFP+ with GFP− myofibers from 6 mice were calculated. As shown in Fig 7D, the muscle fibers with down-regulated Tfam expression were much smaller than the control group, indicative of muscle atrophy. Consistently, the grip strength of AAV-injected hindlimbs was significantly weaker than the contralateral ones with PBS injection, no matter we injected virus at the left or right hindlimb (Fig 7E). Taken together, the iKRAB KI mice provide a versatile model for ex vivo and in vivo LOF studies.

## Discussion

In this study, we generate an inducible CRISPRi mouse ESC line and KI mouse, inducibly and reversibly expressing dCas9-KRAB protein. With either the cell line or animal model, a simple transduction of gRNAs enables us to do any controllable LOF studies.

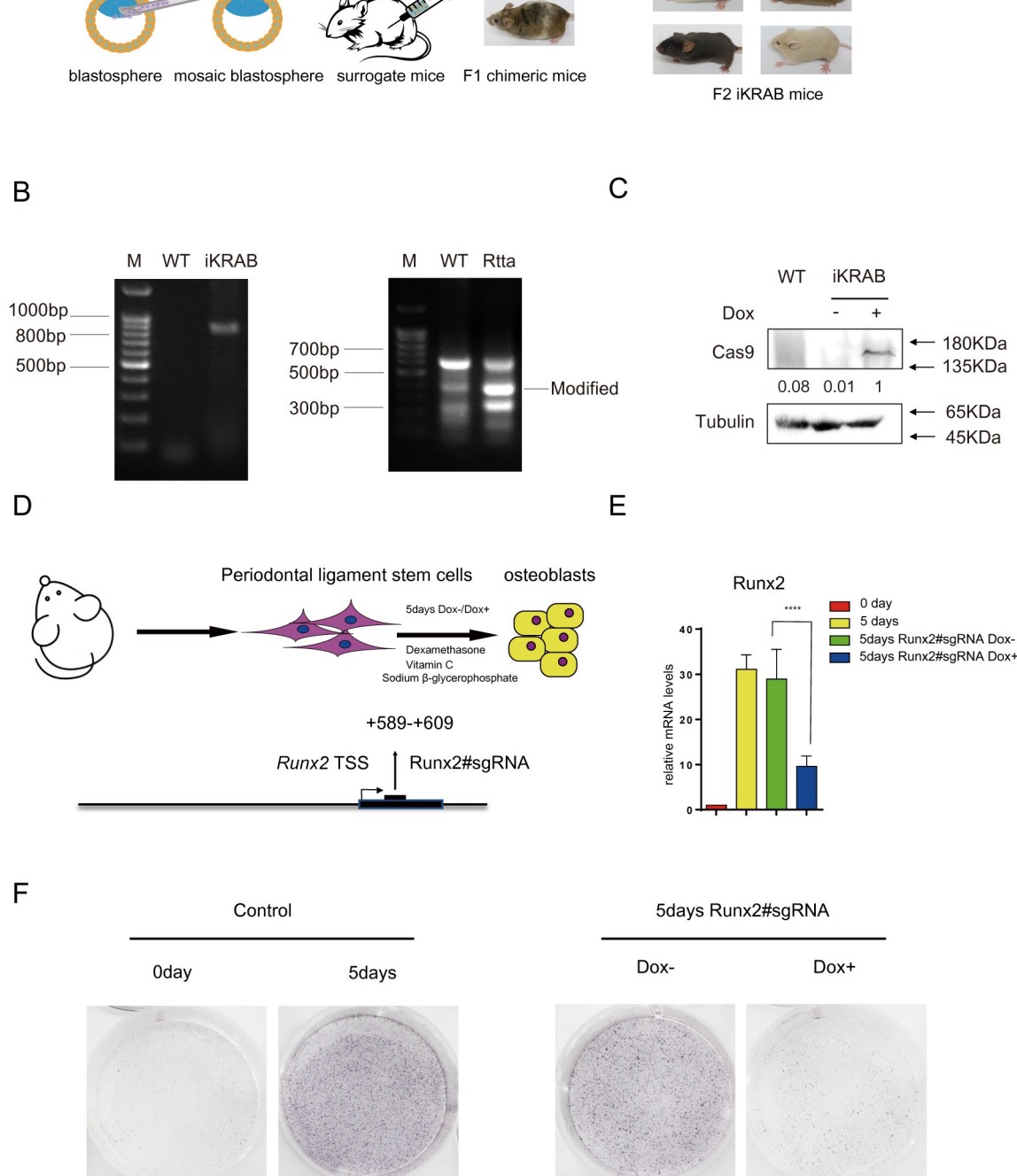

**Fig 6. Characterization of the iKRAB KI mouse and ex vivo effect.** (A) Schematic diagram shows the generation of iKRAB KI mice. (B) Genotyping PCR analysis of TRE-dCas9-KRAB and rtTA in WT and KI mice. (C) Western blot analysis of dCas9-KRAB expression in WT and iKRAB mice. β-tubulin served as a loading control. A relative gray value quantification of dCas9-KRAB protein levels is below each lane of the band. (D) Schematic diagram shows that PDLSCs from the iKRAB KI mice were introduced with sgRNA against the TSS of *Runx2*, followed by differentiation into osteoblasts. (E) RT-qPCR analysis showing Runx2 mRNA levels in the designated groups. Data are represented as the mean ± SD of replicates ($n = 3$) (****$p < 0.0001$; and 2-tailed unpaired $t$ test). (F) ALP staining of cells from the designated groups. The numerical values used to generate graphs in panel E are available in S1 Data. ALP, alkaline phosphatase; KI, knock-in; PLSC, periodontal ligament stem cell; rtTA, reverse transcriptional activator; RT-qPCR, reverse transcription PCR; SD, standard deviation; sgRNA, single-guide RNA; TSS, transcription start site; WT wild type.

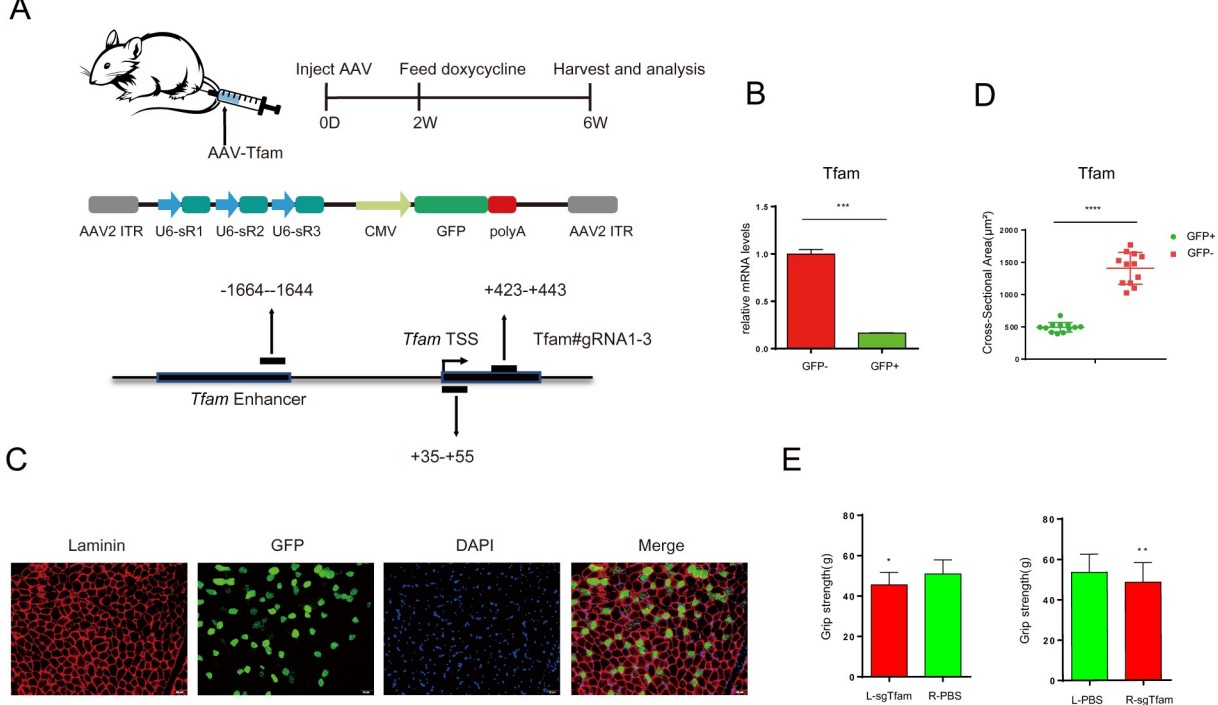

**Fig 7. In vivo inducible CRISPRi effect of the iKRAB KI mouse.** (A) Schematic diagram shows that AAV expressing multiplex gRNAs against Tfam locus was injected into the *tibialis anterior* muscle of the iKRAB KI mice, followed by 1 month of Dox induction and the subsequent analysis. (B) RT-qPCR analysis showing Tfam mRNA levels in the designated groups. (C) IF analysis of laminin expression of the *tibialis anterior* muscle tissue around the injection sites. GFPs mark the infected muscle fibers. The scale bar represents 50 μm. (D, E) Quantification of the cross-sectional area of the *tibialis anterior* muscle fibers and grip strength in designated groups ($n = 6$ mice). L, left; R, right. The data in B, D, and E are presented as means ± SD ($n = 3$ or 6) (*$p < 0.05$, **$p < 0.01$, ***$p < 0.001$, ****$p < 0.0001$; and 2-tailed unpaired *t* test). The numerical values used to generate graphs in panel B, C, and E are available in S1 Data. AAV, adeno-associated viral; CRISPRi, CRISPR interference; GFP, green fluorescent protein; IF, immunofluorescence; KI, knock-in; RT-qPCR, reverse transcription PCR; SD, standard deviation.

CRISPRi techniques have been shown to repress gene transcription to different extent in a variety of cell models [37]. Here, we argued that induced dCas9-KRAB displayed limited effects on active promoters or enhancers. And it was not due to unoptimized gRNA designing because even the same sgRNA achieved differential repressive activities at distinct cellular states (Fig 3). Actually, this issue was also recently raised by several other studies [38–40]. For example, the sgRNA targeting the *MYC* promoter that led to down-regulated MYC expression 6.2-fold in HEK293T cells showed very modest or no repressive activity in cancer cell lines with high levels of MYC expression [39]. This performance is consistent with the transcriptional repression mechanism that KRAB-ZFPs facilitate heterochromatin formation and spreading at inactive chromatin regions [11,13,14,41]. Although dCas9-KRAB protein is tethered to chromatin locally by sgRNAs, it is not sufficient to counteract the active chromatin environment for the propagation of heterochromatin-like features (S3 Fig). And we did find that multi-gRNAs targeting a wide range of regions at the same gene (e.g., *Oct4*, *Nanog*, and *Tfam*) mediate sufficient transcriptional repression (Figs 2B and 7B). Considering the increased risk of off-target of multi-gRNAs, other optimization strategies need to be tested to improve CRISPRi effect on active genes in the future. Recently, KRAB combination with MeCP2 or LSD1 or alternative repressors like SIN3-interacting domain (SID) have been reported to achieve superior efficiency [38,40,42] and are worth further testing at active genes in different cell types.

Despite insufficiency to induce gene inactivation, we showed that dCas9-KRAB preset at inactive promoters or enhancers was sufficient to maintain or foster the inactive state. Taking cell differentiation or reprogramming models, we demonstrated that Dox-induced locus-specific perturbation in iKRAB cells is competent to restrict gene activation and thereby affect cell fate transitions. Moreover, although we only test gRNAs targeting a single gene in our study, we believe that simultaneous introduction of sgRNAs targeting multiple genes would work as well.

In addition to functional studies of individual genes or *cis*-regulatory elements, genome-scale CRISPRi screens have been widely applied to identify genes or noncoding RNAs that control diversity of cellular processes [8,34,38,43,44]. Similarly, we can take iKRAB ESC or its derivatives for CRISPRi screens to identify new genes that regulate stem cell self-renewal and differentiation, to screen potential barriers against reprogramming or to map any key *cis*-regulatory elements especially enhancers for cell fate decisions (Fig 4H). For this purpose, the iKRAB KI mouse will provide a convenient platform for broader applications. Taking adult stem cells or differentiated cells (e.g., fibroblast) from the iKRAB KI mice for ex vivo functional studies or screens will avoid the problems of inadequate ESC differentiation, which will consequently contribute to the improvement of differentiation or reprogramming efficiency for regenerative medicine in the long term. For cellular processes like hematopoiesis, a more physiologically relevant approach is to perform bone marrow transplantation (BMT) after in vitro introduction of gRNAs (Fig 8). There is no doubt that these applications have important implications in developmental and stem cell biology.

More attractively, the iKRAB KI mouse may be valuable for modeling human diseases. Emerging evidences from genome-wide association studies (GWAS) have shown that the vast majority of disease-associated single nucleotide polymorphisms (SNPs) are located in the noncoding genomic regions. However, the causal–effect relationship between these noncoding mutations and phenotypes or diseases could hardly be established until the development of CRISPR-mediated genome and epigenome editing technologies. However, genome editing to generate disease-mimicking cell lines or mouse models that harbor patient-specific noncoding mutations is time-consuming and uneconomical. A trial of epigenome editing in advance to establish the regulatory potential of genomic regions harboring possible causal variants has been suggested [45]. Undoubtedly, our iKRAB systems are at least helpful for testing the conserved LOF variants. A simple delivery of gRNAs in the iKRAB KI mice will accelerate the functional exploration in living organisms that were previously beyond reach. Although our iKRAB KI mouse does not express *Cre* and the tissue-specific CRISPRi cannot be realized so far, the problem can be solved to pre-cross with specific *Cre* mouse. Furthermore, the crossing with existed genetically engineered mouse model (GEMM) will facilitate the modeling of complex human diseases including cancers (Fig 8). Moreover, these animal models will hopefully allow further drug screening and potential preclinical trials. In a word, these versatile iKRAB systems enable a wide range of CRISPRi applications in biological and disease processes.

## Materials and methods

### Ethics statement

All mouse experiments were performed under protocols approved by health guidelines of the Tianjin Medical University Institutional Animal Use and Care Committee in Tianjin, China (Approval number: TMUaMEC-2017009).

### Cell culture

Mouse ESCs were cultured in Glasgow's Minimum Essential Medium (GMEM) with 15% fetal bovine serum (FBS) and 100 U/ml LIF in gelatin-coated plates. In order to maintain the naïve

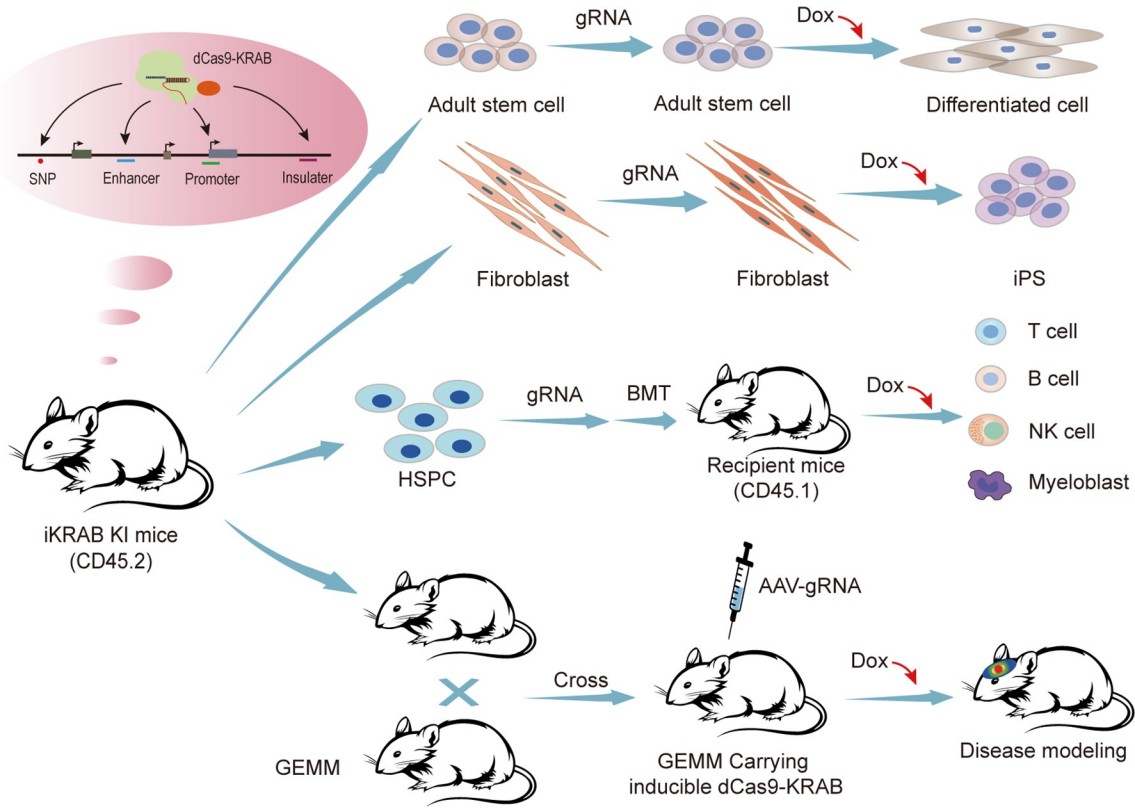

**Fig 8. The versatility of the iKRAB KI mouse.** Primary cells isolated from the iKRAB KI mice, after being transduced by gRNAs in vitro, can be used for following ex vivo or in vivo functional studies or screens. Crossing the iKRAB KI mouse with other GEMM, we can deliver gRNAs targeting conserved regions harboring human SNPs for complex disease modeling. BMT, bone marrow transplantation; GEMM, genetically engineered mouse model; KI, knock-in; SNP, single nucleotide polymorphism.

state, mouse ESCs were cultured in serum-free medium with N2 and B27 supplements, LIF, MEK inhibitor PD0325901 (1 μM), and GSK3 inhibitor CHIR99021 (3 μM). To switch to the primed state, ESC medium was switched to serum-free medium with N2 and B27 supplements, plus Fgf2 (12 ng/ml) and activin A (20 ng/ml) as described [28].

## Cloning and plasmid preparation

The fragment of dCas9-KRAB was amplified from the plasmid (Addgene #50917) and introduced into Gateway Entry vector pCR8/GW/TOPO (Invitrogen, United States) following the manufacturer's protocol and verified by sequencing. The right donor was subcloned into the destination vector p2Lox-FLAG [18] by Gateway Technology (Invitrogen).

## gRNA design and cloning

To minimize off-targets, gRNAs were designed at the following website: http://crispor.tefor. net/. Synthesized oligonucleotides were annealed and cloned into pLX-sgRNA vector following protocol from Addgene (#50662). The sequences of all the gRNAs are listed in S2 Table.

## Generation of iKRAB ESC line

After 16 h of Dox treatment, A2Loxcre mouse ESCs [17] were transfected with p2Lox-FLAG-dCas9-KRAB plasmids using Lipofectamine 3000 (Invitrogen), followed by G418 selection

(50 μg/ml) for 7 days. Individual colonies were isolated after approximately 10 days, expanded, and screened by PCR for inserted sequence. PCR primers are F: TTACCACTCCCTATCAGT GATAG; R: AGGAAGCTCTCTTCCAGCCTATG. And the inducible expression of FLAG-dCas9-KRAB protein was confirmed by western blot assay.

## RT-qPCR

Total RNA was extracted with TRIZOL (Invitrogen). Reverse transcription and quantitative real-time PCR were performed as described [46]. Gene expression was determined relative to *RPLPO* or *Gapdh* using the ΔCt method. The primers are listed in S3 Table.

## ChIP-qPCR

Chromatin preparation was performed as previously described [47]. Briefly, after crosslinking with 1% formaldehyde for 10 min at room temperature and then quenching with 0.125 M glycine for another 5 min, cells were washed with PBS and lysed in SDS buffer (100 mM NaCl, 50 mM Tris-Cl pH 8.1, 5 mM EDTA pH 8.0, 0.5% SDS, protease inhibitors). Nuclei sresuspended in appropriate volume of ice-cold IP buffer (100 mM NaCl, 50 mM Tris-Cl pH 8.1, 5 mM EDTA pH 8.0, 0.3% SDS, 1.0% Triton X-100) was sonicated using a BioRuptor sonicator (Diagenode, Liege, Belgium), followed by centrifugation at 16,000×g for 20 min at 4˚C. Chromatin was then divided in different aliquots that were incubated overnight at 4˚C with primary antibodies. Next, 30 μl protein G magnetic beads were incubated with the reaction for 3 h at 4˚C. Beads were washed 3 times with high salt buffer (1% Triton X-100, 0.1% SDS, 500 mM NaCl, 2 mM EDTA, pH 8.0, 20 mM Tris-HCl, pH 8.0). After reversal of the crosslinking, ChIP DNA was purified for qPCR analysis. The primers are listed in S4 Table.

## NPC and neuronal differentiation

The differentiation was performed following a previous protocol [48]. Briefly, ESCs were cultured in differentiation medium (GMEM medium, 15% FBS, non-essential amino acids, β-mercaptoethanol, L-glutamine, penicillin/streptomycin, sodium pyruvate) to form EB by hanging drops. After 4 days, EBs were collected and cultured on bacteriological Petri dishes with 5 μM ATRA for another 2 days. Then EBs were digest and seeded on 0.1% gelatin-coated plates for another 2 days with RA. After 8 days NPC stage, cells were cultured in N2 medium (DMEM/F12 medium with 3 mg/ml glucose, 1/100 N2 supplement, 10 ng/ml bFGF, 50 U/ml pen/strep, 1 mM/L-glutamine) for another 4 days without ATRA to generate neurons. The derived neurons were maintained in complete neurobasal medium before proceeding for analysis.

## PDLSC extraction and differentiation

PDLSCs were isolated from 8 weeks iKRAB KI mice. Clipping 4 incisors and the surrounding gingival tissue of mice. Repeated washing with PBS. Tissues were digested in Collagenase I and dispase for 1 h and shaked every 15 min in 37˚C. The reaction was then stopped with the same amount of serum, followed by centrifuge and cell seeding on culture dish with StemRD MSC medium for 10 days. For PDLSC differentiation, PDLSC were cultured in osteogenic culture medium (10 mM sodium b-glycerophosphate, 100 μg/l vitamin C, 10 nM dexamethasone) for 5 days [35]. ALP staining was performed according to the protocol from Beyotime Box.

## Lentivirus and AAV preparation

All lentiviruses were generated as previously described [46]. Briefly, lentiviral backbone expressing single or multiplex gRNAs with pAX8 (packaging) and pCMV-VSVG (envelope) plasmids were co-transfected into 293FT cells. After 48 h, virus supernatants were harvested, filtered, and incubated with iKRAB ESCs or primary cells from the iKRAB KI mice. For AAV2 production, HEK293 cells were transfected with the pAAV2 plasmid expressing gRNAs, helper plasmid pDF6, and PEI Max (Polysciences 24765–2, Shanghai, China). At 72 h posttransfection, the cells were rinsed and pelleted via low-speed centrifugation. Afterward, the viruses were applied to HiTrap heparin columns (GE Biosciences 17-0406-01, Shanghai, China) and washed with a series of salt solutions with increasing molarities. During the final stages, the eluates from the heparin columns were concentrated using Amicon ultra-15 centrifugal filter units (Millipore, Burlington, Massachusetts, US).

## Chromosome conformation capture (3C)

3C assays were performed according to previous reports [49]. Cells were crosslinked with 1% formaldehyde for 10 min and quenched with 0.125 M glycine for 5 min. Fixed cells were resuspended in lysis buffer (50 mM Tris-HCl, pH 7.5, 150 mM NaCl, 0.5% NP-40) for 1 h on ice. Nuclei were resuspended in 0.5 ml of 1× restriction buffer with 0.3% SDS and incubated at 37˚C for 1 h. After that, Triton X-100 was added to a final concentration of 1.8% followed by 20 min incubation at 37˚C with shaking. Afterwards, chromatin was digested with HhaI overnight at 37˚C. Restriction enzymes were inactivated by adding SDS to a final concentration of 1.6% and incubating the mixture for 20 min at 65˚C while shaking. The digested chromatin was diluted 10 times and transferred to new tubes. Then digested chromatin was ligated with 100 U of T4 DNA ligase (with Triton X-100 at a final concentration of 1%) for 8 to 14 h at 16˚C. Ligated chromatins were de-crosslinked with 300 mg of proteinase K and incubated at 65˚C overnight, followed by RNase A treatment for 1 h at 37˚C. DNA was then purified by phenol/chloroform extraction followed by ethanol precipitation and re-suspension in water. 3C-PCR primers were designed in HhaI fragments located within the enhancer and promoter regions of interest. In addition, we also designed a pair of PCR primers to amplify an approximately 200 bp fragment without intervening HhaI sites at the *Sox2* locus, which was used as a loading control. The primers are listed in S5 Table.

## SgRNA library preparation and CRISPRi screening

For each transcript, 3 to 5 sgRNAs were designed using CRISPRseek within 500 bp upstream and downstream of the TSS (including alternative TSS). sgRNA sequences that contained BsmbI restriction sites were excluded. The oligonucleotide library was synthesized, annealed, amplified, and ligated into the linearized pKLV-U6gRNA-EF(BbsI)-PGKbsd2ABFP vector (modified from addgene #62348 by replacing puro with bsd resistance cassette) for lentivirus packaging. The iKRAB ESCs were transduced with pooled lentiviral sgRNA with multiplicity of infection (MOI) < 0.3. After selection of blasticidin (10 μg/ml) for 4 days, the transduced cells proceeded for NPC differentiation. A803467 (8 nM) and Dox (1 μg/ml) were treated at 4 days after differentiation. $3 \times 10^6$ cells each from the input and survived cells were harvested for genomic DNA extraction 8 days after differentiation and selection. The double sgRNA-encoding regions were then amplified by PCR followed by next generation sequencing (NGS) library preparation (Vazyme cat.TD503-01, Nanjing, China) and sequencing on an Illumina Hiseq-2500 (San Diego, US). The amplification primers for the library construction and NGS are summarized in S6 Table)

## Generation of animal models and AAV delivery

ESCs were injected to embryonic day 3.5 mouse blastocysts to obtain the founder mice. Chimeric founder mice were bred with C57BL/6 mice, and offsprings with germline transmission were genotyped for rtTA and TRE-dCas9-KRAB transgenes (primers list in S6 Table) and intercrossed to generate iKRAB heterozygous or homozygous KI mice. Animals were fed standard chow diets with access to drinking water ad libitum while housed under a 12-h light–dark cycle. Six-week-old iKRAB KI mice were injected at multiple sites in the *tibialis anterior* with AAV-Tfam (virus titer: 8.1E12) with 30 μl per mice. Injections were carried out under general anesthesia. After 2 weeks, animals were fed by 5% sucrose water with 1 mg/ml Dox for 1 month. Mice were humanely killed via cervical dislocation, and the muscles were rapidly excised.

## IF

Cells were seeded onto slides followed by different treatments and proceeded for IF analysis as previously described [50]. The primary antibodies were listed in S7 Table. The muscle tissue preparation and IF analysis were performed as described [51]. Muscles isolated from tendon to tendon and covered by optimum cutting temperature (OCT) cryoprotectant (Sakura, US) were rapidly passed in liquid nitrogen-cooled isopentane (VWR) for 1 min and left at −80˚C until processed. Frozen samples were cryosectioned at 8-μm thickness using a Leica CM1860 cryostat (Buffalo, US). Then sections were fixed for 10 min with 4% paraformaldehyde in PBS, washed in PBS, and blocked in a solution consisting of 1% Tween-20, 5% BSA, and PBS for 1 h. Then the sections were incubated in anti-laminin antibodies overnight at 4˚C. After 2 washes with PBS-1% Tween-20, samples were incubated with secondary antibodies (1:200, ZSGB-BIO, Alexa-Fluor-594) in PBS for 2 h, followed by 5 min incubation in DAPI nuclear stain (Life Technologies, US). Images were captured using a DP72 fluorescence microscope (Olympus, Japan).

## Measurement of myofiber cross-sectional area

The cross-sectional area of the myofibers was calculated on section images obtained from the *tibialis anterior* muscles using ImageJ.

## Functional grip strength test

Treated and control mice were tested using a commercial grip strength monitor (Chatillon, UK). Briefly, mice were allowed to grip wire mesh of the apparatus by their left or right hindlimbs. Each mouse pulled gently until they released their grip and was given 3 trials per examining period. The force exerted by each hindlimb was recorded for subsequent statistical analysis.

## Supporting information

**S1 Fig. Confirmation of the iKRAB ESC line.** (**A**) Genotyping PCR analysis of proper integration of dCas9-KRAB at the designed locus. (**B**) Western blot analysis showing the inducible expression of FLAG-dCas9-KRAB protein by different concentration of Dox. Gapdh served as a loading control. A relative gray value quantification of dCas9-KRAB protein levels is below each lane of the band. dCas9, deactivated Cas9; Dox, doxycycline; ESC, embryonic stem cell; KRAB, Krüppel-associated box; PCR, polymerase chain reaction.
(TIF)

**S2 Fig. CRISPRi by sgRNAs is insufficient at active genes.** (**A**) Oct4 expression levels are down-regulated around 50% accompanied with higher rate of Nanog-positive cells. IF staining of Oct4 and Nanog in SL-cultured iKRAB cells containing Oct4#sgRNA-4 treated with or without Dox. The scale bar represents 50 μm. Right panel: (**B**) The relative density of Oct4 and the relative Nanog-positive cell numbers are compared in designated group. Data are represented as the mean ± SD of replicates ($n = 3$). ($^{**}p < 0.01$, $^{***}p < 0.001$; 2-tailed unpaired $t$ test). The numerical values used to generate graphs in panel B are available in S1 Data. CRISPRi, CRISPR interference; Dox, doxycycline; IF, immunofluorescence; SD, standard deviation. (TIF)

**S3 Fig. dCas9-KRAB binds at active and inactive chromatin regions comparably.** (**A**) ChIP-qPCR analysis of dCas9-KRAB guided by sgRNAs targeting around the TSS of *Oct4* and *Fgf5* with Cas9, FLAG, and H3K9me3 antibodies respectively. (**B**) ChIP-qPCR analysis of dCas9-KRAB guided by multi-gRNAs targeting around the TSS of *Oct4* with Cas9 and H3K9me3 antibodies, respectively. Data are represented as the mean ± SD of replicates ($n = 3$). The numerical values are available in S1 Data. ChIP, chromatin immunoprecipitation; dCas9, deactivated Cas9; qPCR, quantitative polymerase chain reaction; SD, standard deviation; sgRNA, single-guide RNA; TSS, transcription start site. (TIF)

**S4 Fig. CRISPRi targeting *Oct4*-PE does not down-regulate Oct4 expression in 2i-cultured ESCs.** RT-qPCR analysis of Oct4 expression in stable iKRAB ESCs (2i condition) containing sgRNA against *Oct4*-PE. Data are represented as the mean ± SD of replicates ($n = 3$). The numerical values are available in S1 Data. CRISPRi, CRISPR interference; ESC, embryonic stem cell; PE, proximal enhancer; RT-qPCR, reverse transcription PCR; SD, standard deviation; sgRNA, single-guide RNA. (TIF)

**S5 Fig. CRISPRi targeting *Oct4*-PE hinders the epigenetic changes induced by medium switch.** ChIP-qPCR analysis of epigenomic alterations at PE of *Oct4* with or without Dox treatment during switch from 2i to SL conditions. Data are represented as the mean ± SD of replicates ($n = 3$) ($^{***}p < 0.001$, $^{**}p < 0.01$, $^{*}p < 0.05$; 2-tailed unpaired $t$ test). The numerical values are available in S1 Data. ChIP, chromatin immunoprecipitation; CRISPRi, CRISPR interference; Dox, doxycycline; PE, proximal enhancer; qPCR, quantitative polymerase chain reaction; SD, standard deviation. (TIF)

**S6 Fig. CRISPRi targeting *Sox2*-PE hinders the RA-induced chromatin interaction.** 3C-PCR analysis of *Sox2* PE in designed groups. The primers tested the interaction between *Oct4*-TSS and *Sox2*-PE served as a negative control. A relative gray value quantification of PCR products is below each lane of the band. ChIP, chromatin immunoprecipitation; PCR, polymerase chain reaction; PE, proximal enhancer; TSS, transcription start site. (TIF)

**S7 Fig. Confirmation of Prmt2 knockdown efficiency.** RT-qPCR analysis of Prmt2 expression in the shPrmt2-transduced cells (shPrmt2-1 and -2) with or without Dox induction. Data are represented as the mean ± SD of replicates ($n = 3$) ($^{***}p < 0.001$; 2-tailed unpaired $t$ test). The numerical values are available in S1 Data. Dox, doxycycline; RT-qPCR, reverse transcription PCR; SD, standard deviation. (TIF)

**S1 Table. List of sgRNAs with log10FC > 1 and $p$ < 0.05.** sgRNA, single-guide RNA.
(XLSX)

**S2 Table. SgRNA sequences.** sgRNA, single-guide RNA.
(DOCX)

**S3 Table. RT-qPCR primers.** RT-qPCR, reverse transcription PCR.
(DOCX)

**S4 Table. ChIP-qPCR primers.** ChIP, chromatin immunoprecipitation; RT-qPCR, reverse transcription PCR.
(DOCX)

**S5 Table. 3C-PCR primers.** PCR, polymerase chain reaction.
(DOCX)

**S6 Table. SgRNA and NGS library construction and genotyping PCR primers.** NGS, next generation sequencing; PCR, polymerase chain reaction; sgRNA, single-guide RNA.
(DOCX)

**S7 Table. Antibodies in this study.**
(DOCX)

**S1 Data. Numerical data used in all the figures.**
(XLSX)

## Acknowledgments

We are very grateful to M. Kybe for providing A2LoxCre mouse ESC. We thank the animal, FACS, and imaging facilities at TMU for technical support.

## Author Contributions

**Conceptualization:** Xudong Wu.

**Data curation:** Xudong Wu.

**Formal analysis:** Rui Li, Xianyou Xia, Xing Wang, Xiaoyu Sun, Xudong Wu.

**Funding acquisition:** Xudong Wu.

**Investigation:** Xudong Wu.

**Methodology:** Rui Li, Xianyou Xia, Xing Wang, Xiaoyu Sun, Zhongye Dai, Dawei Huo, Huimin Zheng, Haiqing Xiong, Aibin He.

**Supervision:** Xudong Wu.

**Writing – original draft:** Rui Li, Xianyou Xia.

**Writing – review & editing:** Xudong Wu.

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
