## [Editor Report · Decision Letter 0]

22 Apr 2020

Dear Dr Wu, 

Thank you for submitting your manuscript entitled "Generation and Validation of Versatile Inducible CRISPRi Embryonic Stem Cell and Mouse Models" for consideration as a Methods and Resources by PLOS Biology.

Your manuscript has now been evaluated by the PLOS Biology editorial staff as well as by an academic editor with relevant expertise and I am writing to let you know that we would like to send your submission out for external peer review.

Please re-submit your manuscript within two working days, i.e. by Apr 24 2020 11:59PM.

Kind regards,

Di Jiang

PLOS Biology

---

## [Decision Letter · Decision Letter 1]

29 May 2020

Dear Dr Wu,

Thank you very much for submitting your manuscript "Generation and Validation of Versatile Inducible CRISPRi Embryonic Stem Cell and Mouse Models" for consideration as a Methods and Resources at PLOS Biology. Your manuscript has been evaluated by the PLOS Biology editors, an Academic Editor with relevant expertise, and by four independent reviewers.

The reviews of your manuscript are appended below. You will see that the reviewers find the work potentially interesting. However, based on their specific comments and following discussion with the academic editor, I regret that we cannot accept the current version of the manuscript for publication. We remain interested in your study and we would be willing to consider resubmission of a comprehensively revised version that thoroughly addresses all the reviewers' comments. We cannot make any decision about publication until we have seen the revised manuscript and your response to the reviewers' comments. Your revised manuscript would be sent for further evaluation by the reviewers.

We appreciate that these requests represent a great deal of extra work, and we are willing to relax our standard revision time to allow you six months to revise your manuscript.We expect to receive your revised manuscript within 6 months.

**IMPORTANT - SUBMITTING YOUR REVISION**

*Resubmission Checklist*

*Published Peer Review*

*PLOS Data Policy*

*Blot and Gel Data Policy*

Sincerely,

Di Jiang

PLOS Biology

REVIEWS:

Reviewer #1: Overall, the authors present a resource that will be useful for the scientific community for basic studies as well as disease modeling in vitro and in vivo. However, much of their data is not quantified and there are logical gaps in some of their conclusions. These issues should be addressed prior to publication. 

Major Comments

From data presented in Fig 2 and Fig 5, it is difficult to conclude that CRISPRi by iKRAB is not efficient on actively transcribed genes. 

-For example, the most efficient window for CRISPRi gRNA design is reportedly the 100nt window downstream of the TSS. For OCT4 and BAP1, while the authors show multiple gRNAs, some are outside this window and therefore less likely to be effective. The authors do identify a gRNA for OCT4 which shows 50% knockdown. Showing a range of knockdown efficiencies for different gRNAs is not convincing for their argument above. 

-If BAP1 triggers cell death as discussed by the authors, could the authors be selecting for cells with reduced BAP1 knockdown in Fig. S2B? 

-In Fig 5E, it appears Runx2 expression is already activated by day 5 of differentiation and gRNA activation significantly inhibits Runx2 expression, which is nice to see in the mouse model but seems to conflict with their conclusions that the CRISPRi system is inefficient for silencing active genes. 

-In the paper by Mandegar et al (Cell Stem Cell, 2016), they are able to largely eliminate NANOG expression in multiple iPSC clones using CRISPRi; how do the authors' reconcile these conclusions? 

A significant amount of data used to draw conclusions is not quantified. A few examples are given below. 

-Fig S2A (reportedly showing homogeneous downregulation of OCT4) is not at all convincing. These data should be quantified (e.g., Western blot analysis). 

-Similarly, other immunostaining figures throughout the manuscript used to draw conclusions (e.g., Fig 2D, 3C, 3G, 4C) should be quantified. 

-In Fig 4C, again the data needs to be quantitative in order to any draw conclusions and in this case, the data underlying their conclusions seems particularly weak. From the single images, it appears AMI-1+ alone (upper right) and A803467+ alone (bottom left) significantly alter cell morphology and/or viability. And it is difficult to tell what's going on in the AMI-1+ and A803467+ condition (bottom right). 

-Fig 3G is insufficient to conclude differentiation defects. Are these fewer neurons? Less MAP2 per neuron?

RT-qPCR analyses: 

-Similar fold changes are met with different conclusions by the authors. For example, all of the comparisons in Fig. 3B are significant (**), but based on the text, we are to conclude that the 2i comparison is not relevant, but the SL and FA conditions are.

-The authors conclude that the switch-on of Sox2 PE was blocked in Fig 3E but with essentially the same fold change and significance values, the authors conclude that OCT4 sgRNA#4 (Fig 2A) was of poor efficiency. 

-In line 186-187, the authors reference a slight upregulation of OCT4 expression as shown by RT-qPCR (Fig 3A and S4) but Fig 3A is a schematic and the RT-qPCR data in Fig S4 shows no change. Should this be Fig 3B? If Fig 3B, why does OCT4 go up significantly when targeted under 2i conditions? 

-It would be helpful to include fold-change for different conditions, as the reader has to compare relative changes across different baselines in the current RT-qPCR graphs.

Minor comments

-Please be consistent with nomenclature (e.g., gRNA and sgRNA used interchangeably). 

-The labeling of Fig 6C, D seems off (lines 289-293) as Fig 6D is never referenced in the text. 

Reviewer #2: The submitted manuscript "Generation and Validation of Versatile Inducible CRISPRi Embryonic Stem Cell and Mouse Models" describes generation of mouse ES cells (iKRAB) and consequently mice containing an inducible dCas9-KRAB construct. The authors find that CRISPRi induction does not repress its target locus well at highly expressed genes referred to as "actively transcribed genes" (eg when targeting the Oct4 promoter). The authors leverage transition states of differentiating iKRAB ES cells and demonstrate that dCas9-KRAB is capable of maintaining a repressed state at promoters or block activation at gene proximal enhancers. Finally, mice expressing inducible dCas9-KRAB are created to test repression ex vivo and in vivo. 

Although this manuscript holds a great premise, this reviewer does not feel that the current manuscript lives up to its claims. The authors have made interesting observations that appear to only apply to a subset of cell states. This manuscript is of interest to the field, but claims about versatility cannot be made. Especially, considering another recent study has developed enCRISPRi and enCRISPRa mice (PMID:31980609) and have demonstrated versatility of their system and shown functionality in several mouse tissues. 

Major concerns:

CRISPRi is a well-established system that has been used for high-throughput screens in many studies. In this study, the authors conclude: "Therefore our data argue against the high CRISPRi efficiency by iKRAB on actively transcribed genes". Did the authors intend to refer to highly expressed genes or any gene that is expressed and hence actively transcribed? This claim needs to be better defined and strengthened by careful evaluation of the effect of iKRAB on genes with various expression levels. This is a necessity to evaluate the versatility of these cells.

The authors then identify very specific developmental stages to evaluate their CRISPRi ES cell line. In these particular situations of cell state transitions, the authors show convincingly that dCas-KRAB induction can maintain repression of inactive genes when targeting promoters or proximal enhancers. While these are interesting observations for iKRAB cells, they only apply to particular cell states and transitions and hence will have limited applications. 

The authors further demonstrate the use of iKRAB cells in a screen targeting promoters of chromatin regulators to identify genes involved in toxicity of sodium channel blockers. Here the authors chose gRNAs targeting gene promoters. This is an interesting experiment again taking advantage of transitioning cells and identifying 19 candidate genes. It is unclear if the experiments were carried with biological replicates as this would be important to demonstrate validity of these findings. More thorough experimental design and validation is needed to support these observations.

When the iKRAB mouse model is used in combination with 3 gRNAs delivered to the muscle, significant reduction of TFAM is observed in vivo. All prior experiments are based on a single gRNA. Perhaps the iKRAB ES cells and mouse model would be more versatile when using multiple gRNAs. Could targeting of multiple guides to one target have a synergistic effect and more efficiently repress "actively transcribed genes"? 

Minor comments:

Authors need to clearly identify all plasmids obtained from other sources (Addgene #) and add reference for each of these plasmids. 

Inducible protein expression was confirmed by Western. There should be some reference to which antibody and protocol was used.

Which antibodies were used for ChIP? Please include company and catalog#

In all figures where applicable, please add sgRNA#s to corresponding position diagram. I assume they are graphed in the same order, but that would make it clearer.

Supplementary Figure S6. I did not see the difference between +/- DOX.

Reviewer #3: The goal of this study was to develop tools to suppress genes in murine ES cells and mice. The authors generated murine ES cells in which dCas9-KRAB was inserted into the Hprt locus under the control of a doxycycline (rtTA)-responsive promoter. They then demonstrate that introduction of sgRNAs targeting inactive, but not active, genes can maintain the target genes in an inactive state. They also demonstrate the utility of the ES cells by performing a high-throughput screen and by using them to create mice harboring the dCas9-KRAB cassette. Lastly, they use the novel mouse model to demonstrate suppression of a target gene in vivo.

Comments

1. A major limitation of the tools developed by the authors is the inability of the dCas9-KRAB transgene to silence active genes. Although the authors cite a study demonstrating a similar phenomenon, they do not acknowledge or discuss the numerous studies in which dCas9-KRAB effectively suppresses actively transcribing genes. For example, using an approach very similar to the authors, Conklin and co-workers created human iPSCs harboring a Dox-regulated dCas9-KRAB and showed that it effectively suppressed Nanog, which was robustly expressed prior to Dox addition (see Figure 2C in Cell Stem Cell 18:541, 2016). Indeed, the development of this technology by the Weissman and Qi labs for use in genome-wide screens depends on the ability of CRISPRi to suppress most target genes, irrespective of their initial transcriptional state. No explanation is provided for the lack of effectiveness in this particular model.

2. Another significant limitation is the lack of any demonstration of target gene specificity. While others have demonstrated relatively high specificity of the dCas9-KRAB system, whether such specificity exists in the ES cells and mice generated by the authors is unclear.

3. The statement on page 14 suggesting that in vivo use of dCas9-KRAB is challenging due to its large size is not supported by recent publications (Sci Reports 9:17312, 2019; Nature Neuroscience 21:447, 2018).

Reviewer #4: This manuscript presents data of using an inducible Crispr repression system to probe gene functions and epigenetic modulator-screening in mouse ES cells (iKRAB ESC), and in the mouse. 

The system is in general not a novel one since many similar studies have been published. They however present some interesting findings. For example, the system suppresses gene expression more effectively for the genes that are not initially expressed. For genes that are highly expressing in ESCs, targeting the promoters or TSS is not effective. This finding is in line with other studies where targeting suitable enhancers is much more effective than TSS or promoters for actively expressing genes. The CRISPRi screening in this study has identified toxicity resistance genes which may provide possible solutions for neuroprotection, which is interesting. 

Since the rtTA is inserted into the Rosa26 locus and the TRE-dCas9-KRAB integrated at the Hprt, it is expected that these two transgenes are segregated into different germline transmission mice from chimeras. The authors need to state clearly how the mice carrying the two transgenes are generated for in vivo study. 

In the mouse experiment, the system is used to target the specific enhancer and the first exon of Mitochondrial transcription factor A (TFAM), whose depletion results in muscle atrophy. The data show that Tfam expression levels are significantly lower in GFP+ (AAV-gRNA-expressing) cells, which also have smaller diameter of muscle fibers, indicative of muscle atrophy. Have the authors performed any functional tests of these mice that have putative muscle atrophy? 

Some details also appear missing. For example, the section describing Figure 6B-C is confusing, and apparently mis-cites figure panels. There is no citation of Figure 6D.

---

## [Decision Letter · Decision Letter 2]

9 Oct 2020

Dear Dr Wu,

Thank you for submitting your revised Methods and Resources paper entitled "Generation and Validation of Versatile Inducible CRISPRi Embryonic Stem Cell and Mouse Models" for publication in PLOS Biology. I have now obtained advice from the original reviewers and have discussed their comments with the Academic Editor. 

Based on the reviews, we will probably accept this manuscript for publication, assuming that you will modify the manuscript to address the remaining points raised by the reviewers. IMPORTANT: Please also make sure to address the Data and other Policy-related requests noted at the end of this email.

We expect to receive your revised manuscript within two weeks. Your revisions should address the specific points made by each reviewer. In addition to the remaining revisions and before we will be able to formally accept your manuscript and consider it "in press", we also need to ensure that your article conforms to our guidelines. A member of our team will be in touch shortly with a set of requests. As we can't proceed until these requirements are met, your swift response will help prevent delays to publication.

- a cover letter that should detail your responses to any editorial requests, if applicable

*Copyediting*

*Published Peer Review History*

*Early Version*

Sincerely,

Roli Roberts

Senior Editor,

rroberts@plos.org,

PLOS Biology

ETHICS STATEMENT:

-- Many thanks for including the full name of the IACUC/ethics committee that reviewed and approved the animal care and use protocol/permit/project license. However, please could you also also include an approval number.

-- Please include the specific national or international regulations/guidelines to which your animal care and use protocol adhered. Please note that institutional or accreditation organization guidelines (such as AAALAC) do not meet this requirement.

DATA POLICY:

Regardless of the method selected, please ensure that you provide the individual numerical values that underlie the summary data displayed in the following figure panels as they are essential for readers to assess your analysis and to reproduce it: Figs 2AB, 3BDEG, 4BDFGI, 5BDF, 6E, 7BDE, S2AB, S3, S4,S5, S7. NOTE: the numerical data provided should include all replicates AND the way in which the plotted mean and errors were derived (it should not present only the mean/average values).

REVIEWERS' COMMENTS:

Reviewer #1:

The authors present new data and analysis which overall increase the rigor of the manuscript and the resources they have developed will be useful to the scientific community. However, a few key issues remain which should be addressed prior to publication. 

First, the author's conclusions about dCas9-KRAB exerting modest effects on active gene expression remain confusing and not entirely supported by the data. 

-The manuscript contains a mix of statements like, "after introduction of specific gRNAs, the induced dCas9-KRAB efficiently maintains gene inactivation, though it exerts modest effects on active gene expression" (lines 38-39), "induced dCas9-KRAB displayed limited effects on active promoters or enhancers" (lines 337-338), and "multi-gRNAs targeting a wide-range of regions of the same gene mediate sufficient transcriptional repression" (lines 348-349). Overall, are the authors arguing that dCas9-KRAB has modest effects on active gene expression only when using sgRNAs but that this problem can be resolved by multiplexing? Or that dCas9-KRAB, regardless of single- or multi-gRNAs utilization, has limited effects on active gene expression? This is an important difference and it is not always clear from the text and figures which position the authors are taking. If the authors want to draw conclusions about single- versus multi-gRNAs, then using multi-gRNAs should indeed spread the heterochromatin-like state in contrast to single-gRNAs (Fig S3). However, experiments like this were not done to directly test the hypothesis, if this is the position the authors want to argue. Their conclusions about single- versus multi-gRNAs remain limited; really just data shown for Oct4 for a direct comparison of single- versus multi- gRNA effects on a gene.

-In Fig 4B, there appears to be a rather dramatic effect with an Oct4 PE sgRNA in the SL conditions where the PE is active. Oct4 levels were referred to as "successfully repressed in the SL condition" in line 215 of the text. Was dox treatment given after the change from 2i to SL? Does this conflict with the conclusion of limited effects on active enhances? 

-The point raised previously about BAP1 potentially triggering cell death (and therefore selecting for cells with reduced BAP1 knockdown) means that it is not a great example to select to show limited CRISPRi efficiency. Can the authors show other examples of genes that would not have this confound? 

There are a few concerns with regard to quantification of IF images. 

-Quantification was presented as cell number / mm2, which requires uniform plating density of all cultures for accuracy. These types of analyses should anchor the signal relative to the number of DAPI positive cells. 

-In some cases, it's not clear how quantification was performed. For example, in Fig S2, Oct4 looks present but of reduced intensity in some cells in panel A, but the authors appear to take a positive / not positive approach to quantification in panel B. How did they threshold and quantify images like this? Moreover, the conclusion that "Oct4 expression was homogeneously downregulated after Dox treatment" (lines 130-131) seems at odd with quantification showing a nearly three-fold reduction in the number of positive cells (panel B)? 

Finally, I am confused by Figure 3F/G. The authors state in lines 193-195 that "upon being switched back to 2i medium, a subset of the Mll1 knockdown cells regained Oct4 and Nanog expression. In contrast, Oct4 and Nanog-positive cells could hardly be observed in the control group." In Figure 3F/G, doesn't Mll1 knockdown (+dox) in 2i shows no Oct4 or Nanog expression in contrast to the control or -dox condition? 

Reviewer #2:

The authors have addressed all concerns.

Reviewer #3:

The responses are adequate except for the response to comment 3. The authors claim that in vivo delivery of dCas9 is challenging but still do not adequately support this contention. They mistakenly claim that the references I cited use "super high titer of virus". The definition of "super high" is debatable. However, one of the publications I cited does not use virus at all but instead used a transgene via pronuclear injection, a procedure that is not really challenging at all. I maintain that the authors should modify this statement without providing evidence to support it.

---

## [Editor Report · Decision Letter 3]

2 Nov 2020

Dear Dr Wu,

On behalf of my colleagues and the Academic Editor, Bon-Kyoung Koo, I am pleased to inform you that we will be delighted to publish your Methods and Resources in PLOS Biology. 

PRODUCTION PROCESS

Before publication you will see the copyedited word document (within 5 business days) and a PDF proof shortly after that. The copyeditor will be in touch shortly before sending you the copyedited Word document. We will make some revisions at copyediting stage to conform to our general style, and for clarification. When you receive this version you should check and revise it very carefully, including figures, tables, references, and supporting information, because corrections at the next stage (proofs) will be strictly limited to (1) errors in author names or affiliations, (2) errors of scientific fact that would cause misunderstandings to readers, and (3) printer's (introduced) errors. Please return the copyedited file within 2 business days in order to ensure timely delivery of the PDF proof. 

If you are likely to be away when either this document or the proof is sent, please ensure we have contact information of a second person, as we will need you to respond quickly at each point. Given the disruptions resulting from the ongoing COVID-19 pandemic, there may be delays in the production process. We apologise in advance for any inconvenience caused and will do our best to minimize impact as far as possible.

EARLY VERSION

PRESS 

Kind regards,

Alice Musson

Publishing Editor, 

PLOS Biology

on behalf of

Roland Roberts,

Senior Editor

PLOS Biology